# A Novel Predictive Multi-Marker Test for the Pre-Surgical Identification of Ovarian Cancer

**DOI:** 10.3390/cancers15215267

**Published:** 2023-11-02

**Authors:** Andrew N. Stephens, Simon J. Hobbs, Sung-Woon Kang, Maree Bilandzic, Adam Rainczuk, Martin K. Oehler, Tom W. Jobling, Magdalena Plebanski, Richard Allman

**Affiliations:** 1Hudson Institute of Medical Research, Clayton 3168, Australia; kansu353@gmail.com (S.-W.K.); maree.bilandzic@hudson.org.au (M.B.); adam.rainczuk@bruker.com (A.R.); 2Department of Molecular and Translational Sciences, Monash University, Clayton 3168, Australia; 3Cleo Diagnostics Ltd., Melbourne 3000, Australia; simon.hobbs@cleodx.com (S.J.H.); richard.allman@cleodx.com (R.A.); 4Bruker Pty Ltd., Preston 3072, Australia; 5Department of Gynecological Oncology, Royal Adelaide Hospital, Adelaide 5000, Australia; oehler.mk@gmail.com; 6Robinson Institute, University of Adelaide, Adelaide 5000, Australia; 7Department of Gynecology Oncology, Monash Medical Centre, Bentleigh East 3165, Australia; tjobling@bigpond.net.au; 8School of Health and Biomedical Sciences, RMIT University, Bundoora 3083, Australia; magdalena.plebanski@rmit.edu.au

**Keywords:** ovarian, cancer, CXCL10, biomarker, diagnostic, triage, malignant, benign

## Abstract

**Simple Summary:**

Ovarian cancer remains one of the most lethal malignancies for women, with a complex presentation and, typically, a late-stage diagnosis. Many common benign gynecological diseases can present with similar symptoms to malignancy, and exploratory surgery is required before a conclusive diagnosis can be made. We have developed a new biomarker panel to assist in pre-surgical diagnosis and improve the clinical decision-making process. In a retrospectively collected cohort of 334 women, a multi-biomarker panel measured in plasma correctly identified malignant from benign samples with 95% sensitivity/specificity and out-performed current clinical methods. This new panel may provide a useful clinical adjunct to improve clinical workflows for patients with suspected ovarian malignancy.

**Abstract:**

Ovarian cancer remains the most lethal of gynecological malignancies, with the 5-year survival below 50%. Currently there is no simple and effective pre-surgical diagnosis or triage for patients with malignancy, particularly those with early-stage or low-volume tumors. Recently we discovered that CXCL10 can be processed to an inactive form in ovarian cancers and that its measurement has diagnostic significance. In this study we evaluated the addition of processed CXCL10 to a biomarker panel for the discrimination of benign from malignant disease. Multiple biomarkers were measured in retrospectively collected plasma samples (*n* = 334) from patients diagnosed with benign or malignant disease, and a classifier model was developed using CA125, HE4, Il6 and CXCL10 (active and total). The model provided 95% sensitivity/95% specificity for discrimination of benign from malignant disease. Positive predictive performance exceeded that of “gold standard” scoring systems including CA125, RMI and ROMA% and was independent of menopausal status. In addition, 80% of stage I-II cancers in the cohort were correctly identified using the multi-marker scoring system. Our data suggest the multi-marker panel and associated scoring algorithm provides a useful measurement to assist in pre-surgical diagnosis and triage of patients with suspected ovarian cancer.

## 1. Introduction

### Background

Ovarian cancer remains one of the most lethal gynecological malignancies globally, with 314,000 new cases and 207,000 deaths in 2020 [1]. Appropriate surgical referral and initial management strategy is a key indicator of outcome for ovarian cancer patients; in particular, the 5-year survival for patients with advanced-stage disease is improved when cytoreductive surgery is performed by a gynecological oncologist [2]. A key obstacle in developing appropriate triage protocols, however, is the lack of diagnostic certainty. Ovarian cancers are typically asymptomatic or present as potentially benign conditions. Definitive diagnosis usually occurs post-surgically and often following extensive tissue removal. As a result, less than half of cancer patients are appropriately referred to a gynecological oncology specialist for primary surgery [3,4].

Whilst there are no universally adopted guidelines, clinical work-up for a suspected ovarian malignancy to direct referral for surgery typically involves physical examination, transvaginal ultrasound (TVU) and measurement of serum biomarkers including cancer antigen 125 (CA125) and Human Epididymal Protein 4 (HE4) [5,6]. These measurements are commonly used in the calculation of Risk of Malignancy Index (RMI) or Risk of Malignancy Algorithm (ROMA) scores, used to indicate likelihood of malignancy when an ovarian mass is present [7,8]. However, these modalities lack sufficient sensitivity and specificity for consistently reliable identification of malignancy—particularly for early stage, low volume disease—against the background of other benign gynecological conditions [5]. More recently biomarker-based tests (e.g., OVA1™) have received FDA approval for pre-surgical triage; however, these have not been widely adopted to date.

Recently we reported that the measurement of inflammatory cytokines including Interleukin-6 (IL-6) and C-X-C-Motif Chemokine 10 (CXCL10) in blood serum or plasma were able to discriminate between benign and malignant serous epithelial ovarian cancers [9,10]. In the case of CXCL10, this was achieved by evaluating the ratio of active: total CXCL10 in biological samples, termed the “active ratio” [10]. This “active ratio test” improved the identification of malignancy in a small retrospective patient cohort, particularly when combined with the measurement of CA125. Importantly, CXCL10 measurement was largely independent of stage, suggesting that it could provide a useful addition to standard testing workflows to improve triage for surgical staging of patients with early, low-volume cancers [10].

In this study we report the use of a multi-marker panel, including the measurement of IL6 and the CXCL10 active ratio, for identification and differentiation of benign from malignant tumors. Our data suggest this biomarker panel provides improved differentiation of benign from malignant disease compared to CA125, ROMA or RMI, and can provide a useful measurement for the pre-surgical triage of patients diagnosed with an adnexal mass.

## 2. Materials and Methods

### 2.1. Reagents

Antibodies against intact and total CXCL10, full-length CXCL10 protein standard and all reagents were as previously described [10]. Luminex magnetic bead assay kits (IL-6, HE4) were from Thermo Fisher (cat# RDSLXSAHM05). All other reagents were of analytical grade.

### 2.2. Clinical Samples

Assays were performed on retrospectively collected EDTA-chelated plasma samples accessed from the OCRF-sponsored Ovarian Cancer Tissue Bank, housed at the Hudson Institute of Medical Research, Australia. Ethical approval was obtained from the Southern Health Human Research Ethics Committee (HREC #06032C, #02031B), with all participants providing prior informed written consent. All samples were collected from anaesthetized, chemo-naïve patients who underwent surgery for suspected gynecological malignancies. Patients were excluded if they were <18 years at the time of surgery; had a recent (<2 years) history of breast, ovarian, uterine or other gynecological cancer; had undergone chemo, radio- or immune-therapy within the preceding 12 months; or were immunocompromised at the time of diagnosis. Histological assessment of tumor type, stage and grade, pre-surgical clinical markers (CA125, CEA, CA15.3 and CA19.9), pre-surgical pelvic imaging, age, self-reported menopausal status, pre-existing medical conditions and any prior history of malignancy were obtained from de-identified patient medical records. The imaging data were reviewed and scored according to the RMI2 schedule [11] by a gynecological oncology specialist. Details of the cohort are provided in Appendix A.

The Risk of Malignancy Algorithm (ROMA) was calculated according to [12] using the following formula:Pre-menopausal Predictive Index (PI): −12.0 + 2.38 × Ln(HE4) + 0.0626 × Ln(CA125)
Post-menopausal Predictive index (PI): −8.09 + 1.04 × Ln(HE4) + 0.732 × Ln(CA125)
Predicted Probability (PP): ROMA % = exp(PI)/[1 + exp(PI)] × 100

The Risk of Malignancy Index (RMI) was calculated according to [13] using the following formula:RMI = ultrasound score × menopausal status × serum CA125
where ultrasound score is 1 or 4, menopausal status is 1 (pre) or 4 (post) and serum CA125 is in units/mL. The RMI2 scoring system was used as recommended [11].

### 2.3. ELISA

Biomarker measurements by ELISA were performed using EDTA-chelated plasma samples recovered from previously bio-banked specimens. Plasma samples were thawed on ice and clarified by centrifugation (16,000× *g*, 10 min at room temp) prior to use. The CXCL10 active ratio ELISA was carried out as previously described [10]. Magnetic bead immunoassay for IL-6 and HE4 was carried out as previously described [9]. A total of 100 beads per analyte were counted and the median fluorescence intensity was determined. Quantitation was performed against a standard curve for each analyte using five-parameter logistic curve fitting.

### 2.4. Statistical Analyses

Analyses were performed in Python v3.9.7 (accessed on 1 September 2023; https://www.python.org/) (Python Software Foundation, Wilmington, DE, USA), with Confidence Intervals (CIs) calculated using R v4.3.1 (accessed on 1 September 2023; https://www.r-project.org/) (R Foundation for Statistical Computing, Vienna, Austria). CIs for ROC curves were calculated as described by [14] using the ‘pROC’ package (version 1.18.2) [15] and with all other CIs using the Exact Binomial method via the built-in ‘stats’ library. Plots were generated using GraphPad Prism (v10.0.2 v232) (GraphPad Software, La Jolla, CA, USA).

Exploration and estimation of diagnostic performance for the multi-marker panel (combining CXCL10 active ratio, IL-6, CA125 and HE4) was as follows. Missing values for HE4 and IL-6 in a single sample were assigned as their respective data median in each case. Biomarker measurements were transformed using the Yeo–Johnson method [16], and analytes contributing the greatest linear separation between groups were identified by linear discriminant analysis. A classification model was then defined by fitting a multivariate logistic regression model to the Yeo–Johnson transformed data. Model performance was estimated using repeated stratified K-fold cross-validation (4 folds × 5 repeats), with performance estimated using the mean and standard deviation of AUC across all sub-models. The model was then refit to the entire dataset of *n* = 334 cases, and its performance was re-evaluated. A scoring cutoff point was chosen according to Youden’s J index [17]. Final estimates for the full-dataset model were compared to the cross-validation estimates to assess potential overfit in the model.

For the other classifiers (RMI, CA125 and ROMA), samples with missing values were first removed from the dataset prior to estimation of performance (total numbers remaining CA125, *n* = 334; RMI, *n* = 169; and ROMA, *n* = 333). The calculation of RMI ultrasound information was only recovered for 169 samples. Defined cutoff values for CA125 (≥35 U/mL [18]), RMI (≥200 [7]) and ROMA (pre-menopausal ≥13.1%; post-menopausal ≥27.7% [13]) were from the published literature.

## 3. Results

### 3.1. Patient Characteristics

The characteristics of the retrospective cohort used for testing are provided in Table 1 and Appendix A. A total of 334 patient samples met the requirements of this study and were included for analysis. All patients were recruited following referral to a gynecological oncologist for exploratory surgery. In total there were 164 ovarian malignancies (49%) and 170 benign (51%) cases included, with 34% or 66% from pre- or post-menopausal women, respectively. Malignancy was more common in post-menopausal women (~60% of samples) compared to pre-menopausal (~28% of samples), and diagnosed malignancies were almost exclusively high grade (grade 2–3) ovarian cancers of serous epithelial pathology. Amongst the samples included were 17 (~10%) stage I ovarian cancers, of which 14 (82%) were grade 2–3. Approximately 37% of patients (123/334) had known genetic abnormalities at the time of diagnosis, with relatively even distributions between pre-menopausal (~54%) and post-menopausal (~47%) cohorts. Approximately 50% of the cohort had unknown mutational status.

### 3.2. Individual Marker Performance

Median values for each biomarker according to disease (benign or malignant) are provided in Table 2. Unsurprisingly, significant differences were observed between benign and malignant samples for patient age at diagnosis and all individual markers measured (Figure 1). Each of CA125, HE4 and IL-6 increased in a stage-specific manner and were highest in late-stage (stages 2–4) disease. As previously reported [10] the CXCL10 active ratio was independent of cancer stage (Figure 1A) suggesting it may assist in the differentiation of benign disease from early-stage malignancy. RMI2 score also appeared independent of stage; however, only a limited number of stage I samples (*n* = 7) could be included for analysis due to the absence of imaging data.

Linear discriminant analysis (LDA) was used to identify which individual markers contributed the greatest individual separation between benign and malignant samples. Amongst all biomarkers evaluated, discriminant coefficients with the largest magnitude were, in descending order, HE4, CA125, CXCL10 active ratio and IL-6 (Figure 1B). Total CXCL10 also provided discrimination but was excluded due to collinearity with CXCL10 active ratio.

### 3.3. Development of a Combined Biomarker Model

Commencing with individual biomarker measurements, we evaluated different regression models and marker combinations as follows. The data were first transformed to approximate a standard normal distribution using a Yeo–Johnson transformation [16], and a transformation parameter λ was determined for each biomarker. The transform for any individual marker *x*_i_ was defined by the following;
$${x_{i}}^{\left(\lambda \right)}=\left\{\begin{array}{r} ({\left(x_{i}+1\right)}^{\lambda }-1)/\lambda ,\;\mathrm{if}\;\lambda \neq 0,\;x\geq 0\\ \ln(x_{i}+1),\;\mathrm{if}\;\lambda =0,\;x\geq 0\\ -{\left((-x_{i}+1\right)}^{\left(2-\lambda \right)}-1)/(2-\lambda ),\;\mathrm{if}\;\lambda \neq 2,\;x<0\\ -\ln(-x_{i}+1),\;\mathrm{if}\;\lambda =2,\;x<0 \end{array}\right.$$

The transformed biomarker values were standardized according to *x*^(S)^ = (*x*^(λ)^ − μ)/σ, where *x*^(S)^ is the standardized individual measurement in each case. Model selection for combined biomarker analyses was then estimated using multiple model types (including support vector classifier, decision tree, naive bayes and logistic regression), with performance assessed by repeated stratified k-fold cross-validation (4 folds × 5 repeats). Combinations of up to five biomarkers were analyzed. Primary metrics used for comparison were mean AUC ± SD across the 20 sub-models within each cross-validation estimate. A final linear regression model combining four biomarkers (HE4, CA125, IL-6 and CXCL10 active ratio) was chosen, that provided the highest AUC-SD.

The multivariate logistic regression model was defined as described below and then fit to the transformed and scaled dataset using Maximum Likelihood Estimation. This results in the determination of a set of coefficients β.
Logit (*p*(*x*)) = β_0_ + β_1_*x*^(*s*)^*_HE4_* + β_2_*x*^(*s*)^*_CA125_* + β_3_*x*^(*s*)^*_CAR_* + β_4_*x*^(*s*)^*_IL6_*
where *p*(*x*) indicates probability of malignancy.

Prediction of malignancy for any given observation was then obtained by applying the cutoff to the risk score S, calculated according to S = 10*p*(*x*);
*S* = 10 logit^−1^ = (β_0_ + β_1_*x*^(*s*)^*_HE4_* + β_2_*x*^(*s*)^*_CA125_* + β_3_*x*^(*s*)^*_CAR_* + β_4_*x*^(*s*)^*_IL6_*)

Using Youden’s J index [17], an optimal risk score cutoff point of 3.684 was determined.

Potential overfit was assessed by comparison between cross-validation and full model performance estimates. Good agreement was observed for all metrics (within <1% variation in every case—Table 3) indicating an acceptably low level of overfit and suggesting that performance estimates from the final model were reliable.

### 3.4. A Multi-Marker Panel Out-Performs CA125, RMI and ROMA for the Differentiation of Benign from Malignant Disease

Model performance for discrimination between benign and malignant samples was then assessed, with comparisons of the multi-marker panel score made against standard cutoff values for (≥35 U/mL [18]), RMI (≥200 [7]) and ROMA (pre-menopausal ≥13.1%; post-menopausal ≥27.7% [13]). Metrics for comparison included the area under the curve (AUC), sensitivity/specificity and negative/positive predictive values (Table 4 and Figure 2). Receiver operator characteristic (ROC) curves were generated for each of the multi-marker panel, CA125, RMI and ROMA tests (Figure 2A). The multi-marker panel achieved a clear increase in overall efficacy, providing improved sensitivity/specificity characteristics for differentiation of between benign from malignant samples compared to CA125, RMI2 or ROMA (Figure 2A).

The AUC, sensitivity/specificity, PPV/NPV and overall accuracy for each test were determined for the combined cohort, as well as separately for pre- and post-menopausal samples (Table 4). Overall AUC in the combined cohort was above 0.95 in every case, with the highest AUC (0.98) achieved using the multi-marker panel (Table 4). Sensitivities were similar in each case; however, the specificities of CA125 (0.82) and RMI2 (0.75) were reduced compared to the multi-marker panel and ROMA (Table 4). Corresponding PPV was also comparatively lower for each of CA125 (0.83) and RMI2 (0.66).

When post-menopausal samples were considered separately, a similar pattern was observed; sensitivities ranged from 0.95 (CA125) to 0.99 (multi-marker), whilst specificity again was lower for CA125 (0.84) and RMI2 (0.64). The lowest PPV was again observed for RMI2 (0.70). For pre-menopausal samples considered separately, both multi-marker panel and ROMA actually had reduced sensitivity (0.81 in each case) compared to CA125 and RMI2 (0.91 and 0.89 respectively); however, their specificities substantially exceeded those achieved using either CA125 (0.80) or RMI2 (0.50).

Overall, whilst NPV was similar across each marker score and group (combined, pre- or post-menopausal; between 0.93 and 0.98) the highest PPV was achieved by the multi-marker panel in each test (Table 4). The multi-marker panel was the only test to maintain specificity and PPV values above 90% in every case, demonstrating an overall performance that exceeded that of the other tests. Thus, within this cohort, the multi-marker panel provided improved capability to differentiate benign from malignant disease.

### 3.5. A Multi-Marker Panel Assists in the Identification of Early Stage Cancers

Differentiation of early-stage cancers (FIGO stages I and II) from non-malignant growths is particularly challenging, especially in the case of low-volume cancers where CA125 can be below the ≥35 U/mL threshold [19]. Amongst 17 stage I cancers in the dataset, the multi-marker panel correctly classified 13 (81%) as malignant. These included four stage I cancer samples (two pre- and two post-menopausal) with CA125 < 35 U/mL which were not identified by CA125 or ROMA index; whilst one additional sample (post-menopausal, CA125 = 9 U/mL) was correctly identified by the multi-marker panel. A further four stage II cancers were correctly classified by all scoring systems. RMI2 could not be compared as ultrasound information was not recovered for the majority of these samples.

## 4. Discussion

It is well established that patients diagnosed with OC have significantly improved survival benefit when primary surgery is performed by a specialist gynecological oncology surgeon [20,21]. Appropriate pre-surgical triage is therefore highly desirable in a clinical setting, to ensure cancer patients at all stages derive maximal benefit from treatment [22]. Currently less than 50% of cancer patients receive primary surgical intervention provided by an appropriately trained specialist, due to the difficulties in diagnosis and differentiation from more common benign conditions [4,20,22,23,24]. More effective pre-surgical triage testing is thus required to ensure appropriate referrals occur as early in the clinical workflow as possible.

Our data demonstrates high accuracy using a multi-marker panel incorporating the CXCL10 “active ratio” as a method to discriminate benign from malignant disease. CXCL10 is produced early in cancer progression and can be modified through enzymatic cleavage to produce an inactivated protein [10,25]. We previously demonstrated that the measurement of active and total circulating CXCL10, and calculation of their relative ratio, provided a useful measurement for the differentiation of benign from malignant disease [10]. This study now extends that data in a new cohort of patients, diagnosed with either benign or malignant adnexal mass. By comparison to common clinically used methods including CA125, RMI and ROMA, our multi-marker panel achieved superior sensitivity and specificity for the classification of benign from malignant samples in this dataset. Whilst all tests examined achieved the 75% specificity/80% sensitivity suggested as a minimum requirement for clinical use [13], the multi-marker panel outperformed the other modalities (Figure 2 and Table 4). At a comparative specificity of 95%, the sensitivity of each of CA125 and RMI remained below 80%; whilst ROMA remained under 90% sensitivity at the same threshold. Only the multi-marker panel achieved a 95% sensitivity at this level, highlighting its superior ability to differentiate benign from malignant disease. Moreover, the multi-marker panel operated independently of ultrasound scoring suggesting that a two stage clinical workup—as is currently recommended under American College of Obstetricians and Gynecologists (ACOG) guidelines [26]—should provide a practical improvement in the pre-surgical classification of adnexal masses. Our regression model also did not appear overly affected by menopausal status, suggesting this model may provide broad applicability pre-surgical discrimination of benign from malignant disease.

At present there is no clinically routine pre-surgical method for reliable evaluation and differentiation of benign vs. malignant adnexal mass. Cytoreductive surgery is the cornerstone of cancer management, with complete resection desirable for optimal outcomes [6]. Complete hysterectomy is the norm, with bilateral oophorectomy performed in up to 80% of cases [27]. However, removal of the ovaries predisposes women to multiple co-morbidities including increased risk of cardiovascular disease, dementia and certain cancers amongst others [27,28,29]; current recommendations, therefore, suggest a cautious approach to ovarian removal [30]. In the case of benign disease, which outnumbers malignant diagnoses by ~9:1 [30,31], there is a clear need to differentiate pre-surgically to enhance patient outcomes—particularly in the case of pre-menopausal patients, where fertility preservation may be an important consideration and requires specialist input [32,33,34,35].

In the absence of accepted international guidelines, the most commonly used approaches for clinical work-up of patients with adnexal mass are the Risk of Malignancy Index (RMI) or Risk of Malignancy Algorithm (ROMA) scores [7,8]. Whilst RMI can achieve specificity up to ~95% (i.e., the ability to correctly differentiate between benign and malignant disease), its sensitivity is generally lower at 71–80% (i.e., the ability to correctly identify the presence of disease) [8]. For the detection of stage I-II ovarian cancers, sensitivity is further reduced to ~54% [19]. RMI is also dependent on the quality of ultrasound imaging. ROMA typically exhibits higher sensitivity but lower specificity than RMI [36]; in addition, ROMA cutoffs can differ between suppliers (e.g., pre/post-menopausal values of 11.4%/29.9% Roche Diagnostics; 7.4%/25/3% Abbott Diagnostics) [37]. An alternative, biomarker-based approach is an attractive option for improved identification of malignancy—particularly in the case of early stage (FIGO stage I) and metastatic low volume (FIGO stages II, IIIA1(i) and IIIA2) cancers [38], which can be challenging to correctly identify prior to surgery and often require additional and expensive radiological work-up prior to diagnostic laparotomy. Our multi-marker test correctly identified over 75% of FIGO stage I tumors within the dataset, compared to ~59% using ROMA; and all but four stage II-IV cancers. Accordingly, this multi-marker panel may enhance the rapid and effective triage of patients with early stage and/or low volume tumors to ultimately minimize overall health costs, reduce procedures and time associated with clinical work-up, and maximize treatment outcomes for these patients.

Several biomarker-based tests have been introduced for pre-surgical triage, and currently ROMA™ (Fujirebio, Tokyo, Japan), OVA1™ and OVERA™ (Aspira Women’s Health, Austin, TX, USA) are FDA-approved as aids to assess whether a pre- or post-menopausal woman who presents with a suspicious adnexal mass (reviewed in [17]) is at a low or high likelihood of finding malignancy on surgery. Evaluated independently of CA125, OVA1 achieved a sensitivity of 92% but with a specificity of 35%; in combination with physician’s assessment, a modest increase in sensitivity to 96% was observed [39]. Overall OVA1 was not able to improve on CA125 alone [40]. The OVERA test, an iterative advancement on the original OVA1 test, exhibited improved sensitivity (91–94%)/specificity (69–74%) for the differentiation of benign from malignant adnexal masses [41]. Whilst it is not possible to directly compare our data for each of these biomarker-based tests, the high 95% sensitivity/95% specificity characteristics of our multi-marker panel suggests that it will have utility in the triage of adnexal masses. These findings now require validation using an independent cohort.

The overwhelmingly high mortality of ovarian cancers is in part due to the aggressive nature of the disease coupled with the lack of screening strategies [39,42]. In particular, the 5-year survival for patients diagnosed with early (FIGO stage I) disease is >90%; recurrence rates for these patients are below 20% [43]. Screening is therefore widely believed to be key in reducing mortality from ovarian cancer. Due to the low prevalence of ovarian cancer in the community, a screening test requires a minimum of 99.6% specificity at a minimum sensitivity of 75% [39,42]; currently no testing modalities meet this threshold. Our multi-marker panel achieved a sensitivity of 98.8% at a specificity of 75% in this cohort and correctly identified over 75% of all stage I cancers present, suggesting the potential of these markers to contribute to early-stage detection of ovarian cancers. Further development may present an opportunity to apply new biomarkers such as the CXCL10 active ratio in a future screening context for ovarian cancer.

## 5. Conclusions

Determination of the circulating CXCL10 active ratio contributes positively to the definition of benign from malignant disease and, when incorporated into a four-biomarker-panel, out-performs existing modalities. The assembled biomarker panel and associated scoring algorithm provides a useful measurement to assist in pre-surgical diagnosis and triage of patients with suspected ovarian cancer.

## 6. Patents

Aspects of this study are covered by granted patent 2020404453 and provisional patent 540674PRV.

## Figures and Tables

**Figure 1 cancers-15-05267-f001:**
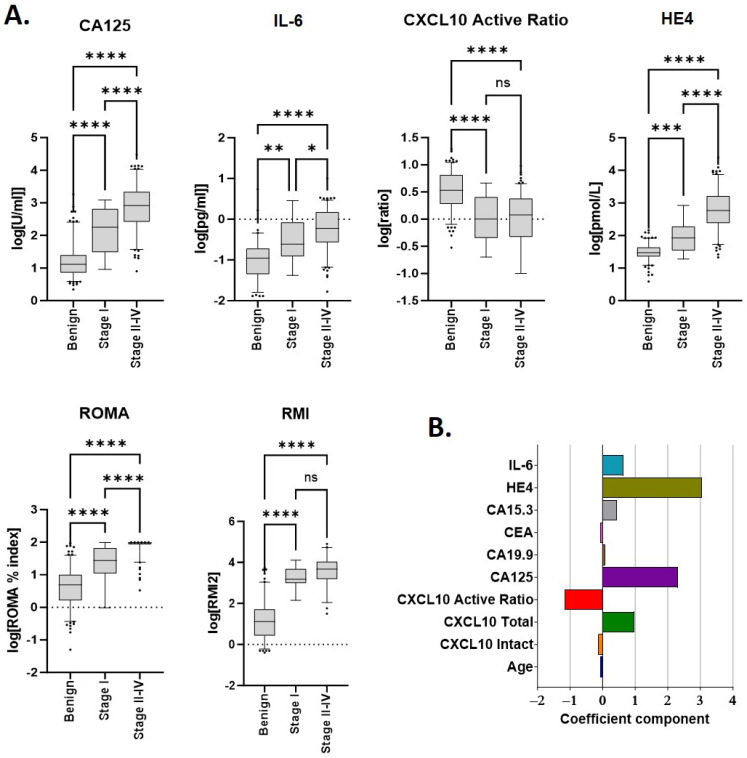
Median marker values and linear discriminants for individual parameters used in scoring. (**A**) Individual analytes and/or calculated scores (ROMA, RMI2) within the cohort, according to disease type and stage. Sample numbers are provided in Table 1 and Table 2. * *p* ≤ 0.01; ** *p* ≤ 0.01; *** *p* ≤ 0.001; and **** *p* ≤ 0.0001. (**B**) Linear discriminant coefficients between groups for each parameter evaluated.

**Figure 2 cancers-15-05267-f002:**
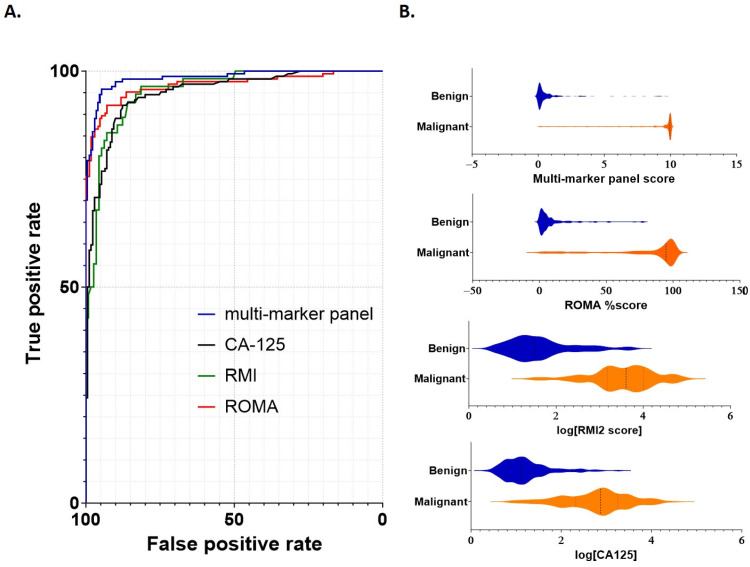
Performance of individual scoring systems for discrimination between benign and malignant disease. (**A**) ROC curves were constructed to assess each scoring system (multi-marker panel, CA125, RMI2 and ROMA). Cutoff values for each marker were as follows: multi-marker panel 3.68; CA125 > 35 U/mL, RMI > 200 and ROMA pre-menopausal >13.1% or post-menopausal >27.7%. (**B**) Violin plots demonstrating comparative scoring across all samples for each scoring system. Sample numbers are as indicated in Table 1 and Table 2.

**Table 1 cancers-15-05267-t001:** Cohort characteristics of patient samples included in this study.

		All	Pre-Menopausal	Post-Menopausal
# Participants (total)		*n* = 334	*n* = 115	*n* = 219
Age at diagnosis (years)	median	65	45	65
	IQ range	47–68	40–49	56–71
Pathology (*n*=)	benign	170	83	87
	malignant	164	32	132
Tumor type (*n*=)	serous	126	18	108
	mucinous	6	3	3
	endometroid	5	1	4
	clear cell	8	4	4
	mixed epithelial	9	3	6
	other	10	3	7
Grade (*n*=)	1	5	2	3
	2	20	8	12
	3	139	22	117
Stage (*n*=)	I	17	11	6
	II	4	0	4
	III–IV	143	21	122
Genetic Predisposition (*n*=)	BRCA1	43	25	18
	BRCA2	49	22	27
	other (lynch, BRIP1+, PALB+, VUS)	31	15	16
	wild type	40	13	27
	unknown	171	40	131
**Ultrasound score (*n*=)**	1	101	50	51
	4	68	16	52
	unavailable	165	49	116

**Table 2 cancers-15-05267-t002:** Individual marker concentrations and calculations.

	Benign		Malignant	
	Biomarker	# Samples	Biomarker	# Samples
	(Median/IQ Range)	(Pre/Post-Menopausal)	(Median/IQ Range)	(Pre/Post-Menopausal)
CA125 (U/mL)	13 (7.2–25.1)	*n* = 83/87	741.5 (210.3–1785.0)	*n* = 32/132
HE4 (pmol/L)	30 (22.7–43.5)	*n* = 83/86	465.6 (193.9–1353)	*n* = 32/132
CXCL10 Active Ratio (pg/pg)	3.4 (1.9–6.5)	*n* = 83/87	1.2 (0.4–2.4)	*n* = 32/132
IL-6 (ng/mL)	0.0 (0.0–0.1)	*n* = 83/86	0.6 (0.2–1.4)	*n* = 32/132
ROMA INDEX % (calculated)	4.2 (1.6–8.7)	*n* = 83/86	94.7 (78.0–100)	*n* = 32/132
RMI score (calculated)	32 (13.1–88.0)	*n* = 57/56	4080 (1487–10,292)	*n* = 10/47
CEA	1.0 (0.6, 1.9)	*n* = 82/87	0.8 (0.5, 1.6)	*n* = 31/121
CA15.3	10.8 (7.1, 14.0)	*n* = 82/87	36.7 (17.3, 89.5)	*n* = 31/121
CA19.9	8.6 (5, 14.6)	*n* = 82/87	9.8 (3.9, 23.1)	*n* = 31/121

**Table 3 cancers-15-05267-t003:** Performance and cross-validation estimates for the multi-marker model.

	AUC	Sensitivity	Specificity	PPV	NPV
**Cross-Validation**	0.981	0.930	0.952	0.950	0.935
**Performance on Full Dataset**	0.984	0.939	0.953	0.951	0.942
**Subtractive Difference %**	−0.26%	−0.85%	−0.13%	−0.09%	−0.64%

**Table 4 cancers-15-05267-t004:** Overall performance metrics for classification of benign vs. malignant disease.

Predictor	Published Cutoff	Menopausal Status	*n*=	AUC (95% CI)	Sensitivity (95% CI)	Specificity (95% CI)	PPV (95% CI)	NPV (95% CI)
Multimarker Panel		combined	334	0.98 (0.97–1.00)	0.95 (0.91–0.98)	0.95 (0.90–0.98)	0.95 (0.90–0.98)	0.95 (0.91–0.98)
	n/a	pre	115	0.95 (0.91–1.00)	0.81 (0.64–0.93)	0.98 (0.92–1.00)	0.93 (0.77–0.99)	0.93 (0.86–0.97)
		post	219	0.99 (0.98–1.00)	0.99 (0.95–1.00)	0.92 (0.84–0.97)	0.95 (0.90–0.98)	0.98 (0.92–1.00)
CA125		combined	334	0.95 (0.93–0.97)	0.94 (0.89–0.97)	0.82 (0.75–0.87)	0.83 (0.77–0.88)	0.93 (0.88–0.97)
	35	pre	115	0.92 (0.86–0.98)	0.91 (0.75–0.98)	0.80 (0.69–0.88)	0.63 (0.48–0.77)	0.96 (0.88–0.99)
		post	219	0.96 (0.93–0.98)	0.95 (0.89–0.98)	0.84 (0.75–0.91)	0.90 (0.84–0.94)	0.91 (0.83–0.96)
ROMA%		combined	333	0.97 (0.95–0.99)	0.93 (0.88–0.96)	0.92 (0.87–0.96)	0.92 (0.87–0.96)	0.93 (0.88–0.96)
	13.1	pre	115	0.93 (0.87–0.99)	0.81 (0.64–0.93)	0.95 (0.88–0.99)	0.87 (0.69–0.96)	0.93 (0.85–0.97)
	27.7	post	218	0.98 (0.97–1.00)	0.96 (0.90–0.98)	0.90 (0.81–0.95)	0.93 (0.88–0.97)	0.93 (0.85–0.97)
RMI2		combined	169	0.95 (0.93–0.98)	0.96 (0.88–1.00)	0.75 (0.66–0.83)	0.66 (0.55–0.76)	0.98 (0.92–1.00)
	200	pre	66	0.98 (0.94–1.00)	0.89 (0.52–1.00)	0.86 (0.74–0.94)	0.50 (0.25–0.75)	0.98 (0.89–1.00)
		post	103	0.94 (0.89–0.98)	0.98 (0.89–1.00)	0.64 (0.50–0.77)	0.70 (0.57–0.80)	0.97 (0.86–1.00)

## Data Availability

The datasets used and/or analyzed during the current study are available from the corresponding author upon reasonable request.

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
