# Peer review of "A Novel Predictive Multi-Marker Test for the Pre-Surgical Identification of Ovarian Cancer"

_cancers, 2023, doi:10.3390/cancers15215267_

Round 1
Reviewer 1 Report
Comments and Suggestions for Authors
The authors propose a novel model that tests for evaluation and detection of early-stage ovarian cancers. The multi-marker model developed using patient samples has been well described and explained throughout the manuscript. The multi-marker test proposed achieves a fairly high percentage of sensitivity and specificity in the given cohort.
However, it would be better if the authors improve the quality of the images of the equations (Line 200, 213, 217). Also, the authors should improve the quality of figure 2. It appears to be pixelated, as well.
Author Response
Reviewer 1:
…it would be better if the authors improve the quality of the images of the equations (Line 200, 213, 217). Also, the authors should improve the quality of figure 2. It appears to be pixelated, as well.
Response: New higher resolution versions of the equations have been generated and inserted into the text. High resolution copies of each figure have also been uploaded to the publishers website for use.
Reviewer 2 Report
Comments and Suggestions for Authors
The manuscript is very well written and the authors claim that their multi-marker panel provides improved differentiation of benign from malignant disease and the associated scoring algorithm provide a useful measurement to assist in presurgical diagnosis and triage of patients with a suspected ovarian cancer.
The introduction and method section are well written.
Please provide a decent size of Figure 1A as it is hard to read the text in the figure. Figure 1B can be reduced to compensate for the enlargement of figure 1A.
In order to clearly discrimination between benign and malignant samples Figure 2A needs to be presented in a better to understand how each sample was compared using the comparisons of the multi-marker panel score made against standard cut-off values for multi-marker panel.
The manuscript is well written but lacks the proper representative figures.
Author Response
Reviewer 2:
1: Please provide a decent size of Figure 1A as it is hard to read the text in the figure. Figure 1B can be reduced to compensate for the enlargement of figure 1A.
Response: The layout of Figure 1 has been altered to increase the size of graphs in Fig 1A, and decrease the size of Fig 1B as suggested.
2: Figure 2A needs to be presented in a better to understand how each sample was compared using the comparisons of the multi-marker panel score made against standard cut-off values for multimarker panel.
Response: The ROC curve is a standard method to define and view binary classifier performance, using the specific cut-off values provided. To assist the reader in interpreting the data, we have added the following text to the legend of Figure 2A; Cutoff values for each marker were; multi-marker panel 3.68; CA125 >35U/ml, RMI >200 and ROMA pre-menopausal >13.1% or post-menopausal >27.7%.